# Prevalence, clinical presentation and factors associated with chronic lung disease among children and adolescents living with HIV in Kenya

**Elizabeth Maleche-Obimbo**[1,2]*, **Engi Attia**[3,4], **Fredrick Were**[1], **Walter Jaoko**[5], **Stephen M. Graham**[6]

1 Department of Paediatrics & Child Health, University of Nairobi, Nairobi, Kenya, 2 Division of Paediatrics, Kenyatta National Hospital, Nairobi, Kenya, 3 Division of Pulmonary, Critical Care and Sleep Medicine, University of Washington, Seattle, Washington, United States of America, 4 Department of Global Health, University of Washington, Seattle, Washington, United States of America, 5 Department of Medical Microbiology & Immunology, University of Nairobi, Nairobi, Kenya, 6 Department of Paediatrics, University of Melbourne, Royal Children's Hospital, Melbourne, Australia

* Elizabeth.obimbo@uonbi.ac.ke

**Data Availability Statement:** All relevant data are within the paper and its Supporting Information files.

## Abstract

### Introduction

Children and adolescents with HIV (CAHIV) may experience recurrent and severe respiratory disease and are at risk of residual lung sequelae, and long-term morbidity from chronically damaged lungs. With improved survival due to increased access to effective antiretroviral therapy there is an increasing population of CAHIV who require optimal life-long care. Chronic lung disease in CAHIV is an under-recognised problem in African settings. We sought to determine the prevalence, clinical presentation and factors associated with chronic lung disease (CLD) among CAHIV in Kenya.

### Methods

CAHIV aged ≤19 years in care at a public hospital in Nairobi were enrolled into a longitudinal cohort study. Sociodemographic and clinical information were obtained through interview, medical record review, physical examination and six-minute walk test. CD4 counts and viral load were determined. Enrolment data was analysed to determine baseline sociodemographic and clinical characteristics. Prevalence of CLD defined as presence of ≥2 respiratory symptoms or signs at enrolment was computed. Logistic regression analysis was performed to evaluate for association between various factors and presence or absence of CLD.

### Results

We enrolled 320 CAHIV of median age 13 (IQR 10–16) years, 80 (25%) were <10 years, 46% were female, 31% lived in a one-room house and 51% used polluting cooking fuel. Antiretroviral therapy (ART) was initiated after age five years in 56%, 43% had prior pneumonia

**Funding:** EMO received the award. Grant no. KNH/R&P/23 I/3/11. Funding Institution: Kenyatta National Hospital. URL: https://knh.or.ke. The funders had no role in study design, data collection and analysis, decision to publish, or preparation of the manuscript.

**Competing interests:** The authors have declared that no competing interests exist.

or tuberculosis, 11% had low CD4 count and 79% were virologically suppressed. Common respiratory symptoms and signs were exertional breathlessness (40%), chronic cough (23%), chest problems in the preceding year (24%), tachypnoea (52%), finger clubbing (6%), exercise limitation (59%) and oxygen desaturation during exercise (7%). CLD was present in 82 (26%) participants, and adding the six-minute walk distance <70% of predicted (exercise limitation) identified an additional 28 (9%) CAHIV with CLD. CLD was more common among older teenagers (odds ratio (OR) 1.95), those who had prior TB or pneumonia (OR 2.04), delayed initiation of ART (OR 2.60), cotrimoxazole prophylaxis (OR 3.35) or TB preventive therapy (OR 2.81). CLD was associated with viraemia (OR 2.7), lower quality of life (OR 12.7), small houses (OR 2.05), caregiver having fewer years of education (OR 2.46), outdoor pollution exposure (OR 3.31) and lower use of polluting cooking fuel indoors (OR 0.26). Adjusted analysis revealed CLD to be associated with prior tuberculosis or pneumonia (adjusted OR (aOR) [95%CI] 2.15 [1.18–3.91]), small house (aOR 1.95 [1.02–3.73]), lower use of polluting cooking fuel (aOR 0.35 [0.13–0.94]) and negative impact on health-related quality of life (aOR 6.91 [3.66–13.03]).

## Conclusions

CLD is highly prevalent across the age spectrum of CAHIV, and most are symptomatic with cough or exertional breathlessness. CLD is associated with prior tuberculosis or pneumonia, socio-environmental factors, and lower quality of life. Structured interventions are needed to provide optimal care specific to their needs.

## Introduction

There are an estimated 1.42 million children aged under 15 years living with HIV, of whom 1.0 million live in the Eastern and Southern African regions [1]. Kenya remains among the top ten highest HIV burden countries with a current estimated 1.4 million (1.2–1.6m) people aged 15 years and above, and 83,000 (66,000–100,000) children age <15 years living with HIV [1].

Much progress has been made in resource-limited settings such as Kenya to achieve early HIV diagnosis and treatment in infants. However, there still exist gaps in the prevention of mother to child transmission cascade and new infant infections continue to occur. As immune depletion can progress rapidly in perinatally HIV-infected infants, many are still diagnosed at an advanced stage of HIV disease having already experienced recurrent severe opportunistic infections such as pneumonia or tuberculosis (TB) [2–5]. Once on effective antiretroviral therapy (ART), recurrent and severe infections become relatively infrequent [6–7] and survival greatly improves but there may be residual organ damage [2, 8–11, 18]. The lung is the commonest site of infection in untreated HIV disease, and therefore the organ most vulnerable to damage and long-term sequelae [2, 12–14]. Recognition is relatively recent by clinicians and researchers that chronic lung disease (CLD) is common among individuals living with HIV, including children and adolescents [2, 10, 14–20].

There is lack of standard paediatric diagnostic protocols and structured services specific to CLD in African healthcare settings. Given improved survival of perinatally HIV-infected children beyond early childhood in sub-Saharan Africa, there is a need for a better understanding of the burden, clinical presentation and severity of CLD in this vulnerable population. These insights may be valuable to guide the targeted clinical services necessary to optimise their

health care, minimise morbidity and improve long-term health outcomes. Therefore, we aimed to describe the prevalence and clinical presentation of CLD in a cohort of children and adolescents with HIV (CAHIV) receiving care in a tertiary public hospital in Nairobi, Kenya, and to determine factors associated with CLD that might inform prevention and treatment strategies.

## Methods

### Study population

We conducted an observational longitudinal cohort study in CAHIV receiving care at a public tertiary referral hospital in Nairobi, Kenya. The Kenyatta National Hospital comprehensive HIV care clinic had approximately 500 children and adolescents age <20 years actively in long-term care in 2020. Participants were recruited between February 2020 and September 2021 into the Breathe Poa Study, a longitudinal cohort study on chronic respiratory disease in children and adolescents living with HIV at Kenyatta National Hospital. This paper reports baseline results of the study.

Children and adolescents were eligible for inclusion if they were HIV infected, aged 19 years or below, and actively in care at the hospital's HIV comprehensive care clinic for at least six months. Those on current treatment for TB or who had acute severe illness on the day of screening were excluded but were eligible to join on completion of TB treatment or recovery from acute severe illness. Individuals with a known diagnosis of cardiac disease (with exception of cor pulmonale) were also excluded.

### Clinical data collection

Clinic attendees at the HIV care clinic were approached during their routine HIV clinic visit, screened for eligibility, and all eligible clients were informed about the study and invited to participate. Informed written consent was obtained from older adolescents aged 18 or 19 years, and from the parent/guardian of those aged <18 years, as well as assent from those children and adolescents aged 8–17 years. A unique study identification number was assigned to each enrolled participant, and this unique study identifier was used on all study case records to protect patient confidentiality. A questionnaire was administered to obtain information on socio-demography, indoor air pollution exposure inside the home, outdoor pollution exposures close to the home or school, history of prior pneumonia or TB, prior hospitalisations due to pneumonia or TB, and current and persistent respiratory symptoms. The Chronic Obstructive Pulmonary Disease Assessment Test (CAT) was administered to evaluate impact of respiratory symptoms on their health-related quality of life. Participants underwent physical examination including measurement of weight, height, resting respiratory rate (RR) and heart rate (HR), chest examination and assessment for finger clubbing. Pulse oximetry was used to determine peripheral oxygen saturations (SPO2).

Ambulatory children and adolescents who were able to follow instructions underwent a standardised six-minute walk test (6MWT) outdoors in a flat grassy field of length 30 metres and total distance covered was measured (outdoors due to covid-19 safety precautions) [21]. Respiratory rate, heart rate and SPO2 before and at end of exercise were documented. Children were asked if they felt short of breath, and how fatigued they felt during exercise. Severity of breathlessness was scored between 1 and 5 (1 = not breathless, 2 = mild, 3 = moderate, 4 = severe or 5 = very severe) using the Medical Research Council (MRC) dyspnoea score [22]. Level of exertional fatigue was assessed using Borg face score between 0 (no fatigue) and 10 (extreme fatigue). Those who were severely hypoxic at rest (SPO2 <90%) or unable to follow instructions because they were mentally or physically impaired or very young (<3 years) were

not subjected to exercise testing. A venous blood sample was obtained on the day of enrolment for CD4 count and full blood count. Recent HIV viral load data were obtained from their medical record and if none had been taken within the preceding 6–12 months, a blood sample was taken for viral load assay.

Relevant prior medical data including HIV diagnosis and staging, ART history and prior medical information were abstracted from the electronic or paper hospital medical records of each patient and entered into study case records forms. In the parent study participants were followed up once, and the study follow-up visit was aligned to their routine scheduled HIV care clinic visit. At follow-up visit each enrolled participant underwent clinical evaluation for intercurrent or persistent respiratory symptoms, and exercise testing as part of the parent study.

Indoor exposure to pollutant cooking fuel smoke at home was captured. Outdoor exposure from car exhaust, smoke from combustion activities such as burning rubbish, or from construction dust near home or school was captured. Participants were asked to specify how frequently they were exposed to each pollutant, whether rarely, some days, or daily.

## Clinical definitions

Body mass index (BMI) was computed for each child using weight in kilograms divided by the square of their height in metres. All anthropometric parameters (weight, height and BMI) for each child were converted to Z-scores by comparing to a reference population [23]. To analyse the 6MWT data, we computed the expected walk distance for each participant using the reference equation developed in healthy North African children by Saad et al. [24]. The actual distance walked during the 6MWT for each child or adolescent was compared to the expected distance walked by a healthy child of similar age, weight, height and sex, and percent of expected distance achieved was calculated. Exercise limitation was defined as reduced walk distance <80% of expected, and mild, moderate and severe exercise limitation defined as 70–79%, 60–69% and <60% of expected distance respectively. Resting RR and HR were categorised as high (tachypnoea and tachycardia, respectively) if they were ≥90th centile for age, and very high if it was ≥99th centile for age [25]. Hypoxia at rest was defined as $SPO_2$ of ≤92% when breathing air, and desaturation during exercise as a decline in $SPO_2$ of 3% or more during the exercise test. Level of breathlessness experienced while walking was classified using the MRC dyspnoea score, and level of exertional fatigue experienced was captured using the Borg face score. The prevalence of each specific respiratory symptom and sign suggestive of CLD was computed.

CLD was defined as presence of ≥2 of the following five clinical features: persistent cough, moderate to severe exertional breathlessness (MRC score >2), very high RR at rest (≥ 99th centile for age), finger clubbing, resting SPO2 ≤92% or decline by 3% or more with exercise. Analysis of each child's data was done to identify those with two or more suggestive clinical symptoms/signs compatible with the diagnosis of CLD, and the prevalence of CLD in the study population using this composite definition was determined (basic definition of CLD). We also explored the effect of adding a sixth criteria of moderate to severe exercise limitation to enhance the identification of participants with CLD (enhanced definition of CLD).

## Sample size, data management and statistical analysis

Epi-Info version 7 Statcalc software was used employing the Fleiss formula with correction to estimate required sample size for cohort studies, assuming 80% power and α (two-sided) of 0.05 [26]. An additional 10% was added to cater for possibility of incomplete data or loss to follow-up giving a desired sample size of 320 study subjects. All identified eligible participants

attending clinic every weekday who consented to participate were consecutively enrolled until the desired sample size was achieved. Data were collected on paper case record forms and entered into a secure database using Microsoft Access 2016 software. A link log of patient study identification number and patient hospital number was maintained in a password protected file with access restricted to the study clinical team to enable feedback on laboratory results to the primary clinical care team and to the study participants, and for rescheduling of missed follow-up appointments in the parent study. Data were cleaned and exported to Microsoft Excel, and analyses conducted using Stata/SE version 17.0 software (StataCorp, Texas, USA). Patient sociodemographic, environmental and clinical characteristics were analysed and reported using median and inter-quartile range (IQR), and frequencies and proportions, as appropriate. Inferential statistics were used to evaluate associations between various sociodemographic and clinical factors and the binary outcome presence or absence of CLD (5-clinical feature definition) using univariable logistic regression and odds ratios with 95% confidence intervals reported. Where appropriate chi-square test for homogeneity across multiple categories, or test for linear trend were applied. Mann-Whitney test was used as appropriate to assess association between continuous variables and CLD. A multivariable logistic regression model was fitted including a-priori factors with p<0.05 at univariable analysis. Variables missing data for >10% of participants were excluded from the model to minimise bias. Adjusted odds ratios with 95% confidence intervals were reported.

The protocol was approved by the University of Nairobi / Kenyatta National Hospital Ethical Review Committee (P396/05/2019) and was registered with the Research Department of Kenyatta National Teaching Hospital. Additional information regarding the ethical, cultural, and scientific considerations specific to inclusivity in global research is included in S1 Checklist.

## Results

### Sociodemographic and home environmental characteristics

We screened 326 children and adolescents from February 2020 to September 2021, and 320 consented to participate and were enrolled. Two were ineligible as they were from a children's home, and four potentially eligible children declined to participate due to non-interest. Due to the covid-19 pandemic active enrolment was stopped on 18[th] March 2020 after enrolling the first 14 participants, and restarted eight months later on 25[th] November 2020 with covid-19 precautions in place for participants and staff. All participants completed the interview, with 56 older adolescents responding to the interview questions directly, and for 264 participants their accompanying parent or guardian assisted the child/adolescent with responding to the interview questions. All participants underwent physical examination and pulse oximetry, and 312 completed the six-minute walk exercise test.

The 320 enrolled participants were of median (IQR) age 13.1 (10.1–16.1) years; 80 (25%) were aged <10 years, 147 (46%) were female, and the majority (57%) were in primary school (Table 1). They were largely of lower socio-economic status as reflected by their home characteristics; 61% lived in one- or two-room houses, 71% of homes used pit latrines, with 44% of households using a shared latrine with other households. Pollution exposure was moderate to high from various sources; 51% to polluting cooking fuel at home, 87% to air pollution near home or school and 76% to car exhaust from high vehicle traffic near their home or school. CAT scores varied ranging from 0–22, with 78%, 10.3% and 11.3% participants having scores between 0–4, 5–9 and 10–22 respectively reflecting low, average and higher impact of their respiratory symptoms on their overall health-related quality of life.

Characteristics of the 264 parents/guardians who accompanied the participant were as follows: 4.5% were age <25 years, 21% age 25–35 years, 48% age 36–45 years, and 25% above 45

**Table 1. Sociodemographic and home environmental characteristics of study participants.**

| Characteristic | Detail | Frequency or median | Percent or IQR |
|---|---|---|---|
| **Sociodemography** | | | |
| **Sex** | Female | 147 | 45.9% |
| **Current age (years)** | *Median* | *13.1* | *10.1, 16.1* |
| **Current age group (years)[a]** | <7 | 41 | 12.8% |
| | 7 to 9 | 39 | 12.2% |
| | 10 to 14 | 132 | 41.3% |
| | 15 to <20 | 108 | 33.8% |
| **Current level of education** | Pre-primary | 38 | 12.0% |
| | Primary | 183 | 57.0% |
| | Secondary and higher | 99 | 31.0% |
| **No of rooms in house** | 1 | 99 | 30.9% |
| | 2 | 97 | 30.3% |
| | 3+ | 124 | 38.8% |
| **Household density** | No. persons per room | 2 | (1, 3) |
| **Type of toilet** | Flush | 92 | 28.8% |
| | Pit latrine | 228 | 71.3% |
| **Shared toilet** | Yes | 141 | 44.1% |
| | No | 179 | 55.9% |
| **Air Pollution** | | | |
| **Polluting cooking fuel** | Low | *159* | 49.7% |
| | Moderate | 101 | 31.6% |
| | High | 60 | 18.8% |
| **Car exhaust exposure** | Low | 77 | 24.1% |
| | Moderate | 48 | 15.0% |
| | High | 195 | 60.9% |
| **Outdoor pollution exposure** | Low | 42 | 13.1% |
| | Moderate | 91 | 28.4% |
| | High | 187 | 58.4% |
| **COPD Assessment Test (impact of respiratory disease on quality of life)** | | | |
| **CAT score categories** | 0–4 (low impact) | 251 | 78.4% |
| | 5–9 (average impact) | 33 | 10.3% |
| | 10+ (higher impact) | 36 | 11.3% |
| **Caregiver Characteristics** | | | |
| **Who filled the interview form?** | Parent/guardian | 264 | 82.5% |
| | Adolescent | 56 | 17.5% |
| **Relationship of accompanying caregiver to child** | Mother | 192 | 72.7% |
| *N = 264* | Father | 27 | 10.2% |
| | Other[b] | 45 | 17.0% |
| **Accompanying caregiver age in years** | <25 | 12 | 4.5% |
| *N = 264* | 25–35 | 54 | 20.5% |
| | 36–45 | 126 | 47.7% |
| | >45 | 65 | 24.6% |
| | Not stated | 7 | 2.7% |
| **Caregiver no. of years education[c]** | 0 to 8 years | 95 | 33.7% |
| *N = 282* | 9 to 12 years | 115 | 40.8% |
| | >12 years | 72 | 25.5% |

*(Continued)*

**Table 1.** (Continued)

| Characteristic | Detail | Frequency or median | Percent or IQR |
|---|---|---|---|
| | Median no. years | 12 | 8, 13 |

a-Additional detail on children <10 yrs: 14 (4.4%) were age <5 yr; 66 (20.6%) were age 5–9 yrs.

b-Other: grandparent 11 (4.2%), aunt/uncle 26 (9.8%), elder sibling/cousin 8 (3.0%). C-includes information

on parent/guardian education provided by older adolescents age 18–19 yr who came unaccompanied.

years. Seventy-three percent were mothers, 10% fathers, 4% grandparents, and 10% aunty or uncle to the participant. Regarding their caregiver education 34%, 41% and 25.5% had received between 1–8 years, 9–12 years, and >12 years of formal education respectively.

## HIV clinical characteristics

Median age for ART initiation was 5.5 (IQR 3.5–7.9) years, and 44% had initiated ART before the age of five years (Table 2). At enrolment into HIV care, 46% of 252 had advanced HIV disease (WHO clinical stage III or IV). Overall, 17.2% had history of TB and 33.1% had history of pneumonia. Prophylactic cotrimoxazole was started before age five years in 34% of the participants, and isoniazid preventive therapy was taken prior to age 10 years by only 25%.

Highest recorded viral load during follow-up was available for 301 participants, and 28% had VL >10,000 c/ml. Of note, viral load was not always routinely performed prior to ART initiation due to programmatic guideline changes over the years and therefore these reported values reflect VL after ART initiation for some children. Current median CD4 percent was 31% (IQR 27%, 37%), and current median CD4 count was 897 (IQR 685, 1148) cells/mm$^3$, and 11% of children were immune suppressed with CD4 count <500 cells/mm$^3$. Recent viral load revealed that 79% had undetectable virus (<50 copies/ml), 13% had low viraemia 50–999 copies/ml (c/ml), and 8% had higher viraemia of ≥1000. Current nutritional status was as follows: 9.4% had moderate-to-severe wasting (BMI z-score <-2), 16% were moderate-severely stunted (HAZ <-2), and 13% were moderate-severely underweight (WAZ <-2).

## Clinical presentation of respiratory disease in children and adolescents

**Respiratory symptoms and signs.** Recurrent or persistent symptoms reported in the preceding twelve months include: breathlessness (40%), cough (23%), sputum/congested chest (15%), tight chest (13%) and chest pain (8%) (Table 3). Seventy-eight (24%) had experienced a chest problem in the preceding year, of these, 45 had experienced chest problems two or more times. Respiratory-focused physical examination revealed that 52% of participants had tachypnoea, 6% had tachycardia, 6% had finger clubbing and only 3% had abnormal chest auscultatory findings. No participant had hypoxia at rest, SPO2 ranged from 95% to 99%.

Burden of respiratory symptoms are further reported by age-group in Table 3. The 80 enrolled children age <10 years had slightly higher burden of cough, chest congestion and recurrent chest problems in the preceding year than the 240 enrolled adolescents. Conversely exertional dyspnoea, tight chest and chest pain were reported more often among adolescents than children.

**Six minute walk test (6MWT).** Seventy-three children age <10 years and 239 adolescents successfully completed a 6MWT, results are displayed in Table 3. Of note, 22 children age <6 years successfully completed the exercise test (youngest were 3.1 and 3.7 years) and only two children deviated from brisk walk protocol by skipping for a short distance during the walk.

**Table 2. HIV clinical characteristics of the study participants.**

| Characteristic | Detail | Frequency or median | Percent or IQR |
|---|---|---:|---|
| **HIV Disease Characteristics at entry to care and during follow-up** | | | |
| **WHO stage at enrolment to care** | I—II | 136 | 54.0% |
| N = 252 | III—IV | 116 | 46.0% |
| **Age started ART in yrs[a]** | <2 | 38 | 12.0% |
| N = 317 | 2 to <5 | 102 | 32.2% |
| | 5 to <10 | 126 | 39.7% |
| | 10+ | 51 | 16.1% |
| | *Median* | *5.5* | *3.5, 7.9* |
| **Highest ever viral load (c/ml)[b]** | <10,000 | 217 | 72.0% |
| N = 301 | 10,000 - <100,000 | 44 | 15.0% |
| | 100,000+ | 40 | 13.0% |
| **Prior lung infection** | TB and pneumonia | 22 | 6.9% |
| N = 320 | Pneumonia only | 84 | 26.3% |
| | TB only | 33 | 10.3% |
| | No TB no pneumonia | 169 | 52.8% |
| | Don't know | 12 | 3.8% |
| **Age started cotrimoxazole (yrs)** | <5 | 108 | 34.4% |
| N = 314 | 5 to 9 | 169 | 53.8% |
| | 10+ | 37 | 11.8% |
| **Age took isoniazid prophylaxis (yrs)** | <10 | 69 | 25.0% |
| N = 273 | 10 to 14 | 133 | 49.0% |
| | 15+ | 71 | 26.0% |
| **Current HIV Clinical Characteristics** | | | |
| **Current ART regimen** | First line | 272 | 85.0% |
| N = 320 | Second or 3rd line | 48 | 15.0% |
| **Current CD4 percent** | <15% | 10 | 3.3% |
| n = 306 | 15 to 24% | 48 | 15.7% |
| | 25 to 34% | 142 | 46.4% |
| | 35%+ | 106 | 34.6% |
| **Current CD4 count (cells/mm3)** | <500 | 36 | 11.3% |
| N = 318 | 500–999 | 158 | 49.7% |
| | 1000+ | 124 | 39.0% |
| **Latest viral Load (c/ml)** | <50 | 252 | 78.8% |
| N = 320 | 50–999 | 41 | 12.8% |
| | 1000+ | 27 | 8.4% |
| **Body mass index Z score[c]** | BMI-Z ≥ -1 (normal) | 221 | 68.9% |
| N = 320 | BMI-Z < -1 to -2 (mild) | 69 | 21.5% |
| | BMI-Z < -2 (mod-severe)[f] | 30 | 9.4% |
| **Weight for age Z score[d]** | WAZ ≥ -1 (normal) | 202 | 63.0% |
| N = 320 | WAZ < -1 to -2 (mild) | 76 | 24.0% |
| | WAZ < -2 (mod-severe) | 42 | 13.0% |
| **Height for age Z score[e]** | HAZ ≥ -1 (normal) | 190 | 59.0% |
| N = 320 | HAZ < -1 to -2 (mild) | 80 | 25.0% |
| | HAZ < -2 (mod-severe) | 50 | 16.0% |

a-Yrs denotes years b-c/ml = copies/millilitre c-BMI denotes wasting. d-WAZ denotes underweight. e-HAZ denotes stunting. f-Mod-severe denotes moderate-to-severe.

**Table 3. Burden and spectrum of respiratory symptoms and signs and six-minute walk test results among all CAHIV, and by age category.**

| Clinical Feature (N = 320) | Detail | All Participants (N = 320) | | Children 0–9 yr (N = 80) | | Adolescents 10–19 yr (N = 240) | |
|---|---|---|---|---|---|---|---|
| | | Freq | % | Freq | % | Freq | % |
| **Respiratory Symptoms (N = 320)** | | | | | | | |
| **Cough** | Any cough | 73 | 22.8% | 22 | 27.5% | 51 | 21.3% |
| | *Mod-severe* | 47 | 14.7% | 17 | 21.3% | 30 | 12.5% |
| **Congested chest or sputum** | Any congestion | 49 | 15.3% | 15 | 18.8% | 34 | 14.2% |
| | *Mod-severe* | 37 | 11.6% | 13 | 16.3% | 24 | 10.0% |
| **Breathlessness[a]** | Any breathlessness | 129 | 40.3% | 25 | 31.3% | 104 | 43.3% |
| | *Mod-severe* | 89 | 27.8% | 17 | 21.3% | 72 | 30.0% |
| **Tight chest** | Yes | 40 | 12.5% | 7 | 8.8% | 33 | 13.8% |
| | *Mod-severe* | 25 | 7.8% | 5 | 6.3% | 20 | 8.3% |
| **Chest pain** | Yes | 26 | 8.1% | 0 | 0.0% | 26 | 10.8% |
| **Had a chest problem in past year** | Yes | 78 | 24.4% | 22 | 27.5% | 56 | 23.3% |
| **No episodes of chest problem past year** | One | 33 | 10.3% | 4 | 5.0% | 29 | 12.1% |
| | Two or More | 45 | 14.1% | 18 | 22.5% | 27 | 11.3% |
| **Examination Findings (N = 320)** | | | | | | | |
| **Tachypnoa at rest[b]** | ≥90th centile | 167 | 52.2% | 30 | 37.5% | 137 | 57.1% |
| *RR centile for age* | ≥99th centile | 91 | 28.4% | 19 | 23.8% | 72 | 30.0% |
| **Tachycardia at rest[c]** | ≥90th centile | 20 | 6.3% | 5 | 6.3% | 15 | 6.3% |
| *HR centile for age* | ≥99th centile | 4 | 1.3% | 0 | 0.0% | 4 | 1.7% |
| **Finger clubbing** | Present | 18 | 5.6% | 9 | 11.3% | 9 | 3.8% |
| **6 Minute Walk Test Results (N = 312)** | | **All Participants (N = 312)** | | **Children 0–9 yr (N = 73)** | | **Adolescents 10–19 yr (N = 239)** | |
| **Percent of expected distance achieved** | Median (IQR) | 77 | 83–70 | 74 | 81–69 | 78 | 83–72 |
| | Min, max | 312 | 42–105 | 73 | 46–101 | 239 | 42–105 |
| **Degree of exercise limitation** *(% of expected distance achieved)* | None (≥80%) | 129 | 41.3% | 21 | 28.8% | 108 | 45.2% |
| | Mild (70–79%) | 117 | 37.5% | 32 | 43.8% | 85 | 35.6% |
| | Mod (60–69%) | 53 | 17.0% | 14 | 19.2% | 39 | 16.3% |
| | Severe (50–59%) | 10 | 3.2% | 5 | 6.8% | 5 | 2.1% |
| | V severe (<50%) | 3 | 1.0% | 1 | 1.4% | 2 | 0.8% |
| **Drop in oxygen saturation during exercise[d]** | Drop 0–1% | 249 | 79.8% | 58 | 79.5% | 191 | 79.9% |
| | Drop by 2% | 41 | 13.1% | 9 | 12.3% | 32 | 13.4% |
| | Drop by ≥3% | 22 | 7.1% | 6 | 8.2% | 16 | 6.7% |
| **Breathless during exercise (MRC dyspnoea score)[e]** | 1 (no dyspnoea) | 104 | 33.3% | 34 | 46.6% | 70 | 29.3% |
| | 2 (mild) | 93 | 29.8% | 18 | 24.7% | 75 | 31.4% |
| | 3 (moderate) | 105 | 33.7% | 20 | 27.4% | 85 | 35.6% |
| | 4 (severe) | 10 | 3.2% | 1 | 1.4% | 9 | 3.8% |
| | 5 (very severe) | 0 | 0.0% | 0 | 0.0% | 0 | 0.0% |

Participant report of fatigue during the exercise test using Borg face score (N = 306): no fatigue-27%, mild-37%, moderate-30%, severe fatigue-7%. a-Get out of breath when climbing stairs or a hill. b-Respiratory rate centile for age. c-Heart rate centile for age. d-Difference in oxygen saturation measured by pulse oximetry pre- and post-exercise. e-Medical Research Council dyspnoea score.

Children with severe symptoms (n = 3) too young <3 years (n = 2), mentally impaired (n = 2) or physically disabled (n = 1) did not undertake the exercise test.

The 312 children and adolescents walked a median (IQR) distance of 573 (508–629) metres. The minimum and maximum distance walked were 256 and 863 metres respectively, and

computed expected distance ranged from a low of 42% to a high of 105% compared to the healthy reference population. One hundred and eighty-three (59%) of 312 participants had reduced walk distance (<80% of expected distance). Further categorisation of reduced walk distance was as follows—37.5% achieved between 70–79% of expected (mild exercise limitation), 17% achieved 60–69% expected (moderate limitation), and 4% walked below 60% expected distance (severe limitation). Moderate-to-severe exercise limitation was proportionately more common among children <10 years than adolescents (27% versus 19% respectively).

Assessment of breathlessness at the end of the 6MWT by MRC dyspnoea score revealed that 67% experienced dyspnoea (score 2–4). Severity of dyspnoea was mild in 30%, moderate in 34%, and severe in 3%, with 33% reporting no dyspnoea. Moderate to severe dyspnoea during exercise (MRC 3–4) was more common among adolescents' than children (39% vs 29% respectively). Comparison of pre- and post-exercise SPO2 revealed that 22 (7%) of participants desaturated by three percent or more during exercise. Fatigue during exercise using Borg face score was reported as follows: 27% were not fatigued, 37% felt mild, 30% moderate, and 7% severe fatigue.

## Prevalence and severity of chronic lung disease

**Chronic lung disease–basic definition (≥2 of five clinical criteria).** Synthesis of participant-specific data on respiratory clinical symptoms and signs suggestive of CLD is displayed in Table 4. Of the 320 participants: 37% had no respiratory symptoms or signs, 38% had one, 18% had two, and 7.6% had three or more clinical characteristics suggestive of CLD. A total of 82 participants had two or more of five clinical features, giving a CLD prevalence of 25.6% (95% confidence interval 20.8–30.4%) using this basic definition.

The commonest symptom phenotypes of breathlessness and cough were prevalent as follows among the 82 participants with CLD; 40 (49%) had cough, and 67 (82%) were breathless (MRC score >2); 10 (12%) had cough only, 37 (45%) had breathlessness only, 30 (37%) had both cough and breathlessness, while five participants had neither cough nor breathlessness. Analysing severity of CLD symptoms and signs among the 82 participants with CLD, 17% had severe cough (scored 4–5 on the 5-point Likert scale of the CAT), 11% had severe breathlessness (MRC score 4), 21% experienced a decline in oxygen saturation of 3% or more during exercise testing and 9% had finger clubbing.

**Enhanced definition of chronic lung disease (≥2 of six criteria).** Adding the criterion of moderate-to-severe exercise limitation (6MWT distance <70% of expected distance for age) to the definition, resulted in identification of an additional 28 (total 110) children as having CLD (Table 4). The prevalence of CLD by this enhanced definition increased from 25.6% to 34.4% (95% CI 29.2–39.6%). Of the 110, 39% had cough and 74% were breathless; 13 (12%) had cough only, 52 (47%) had breathlessness only, 30 (27%) had both cough and breathlessness and 15 had neither cough nor breathlessness. Severity of CLD symptoms and signs were as follows: 13% had severe cough, 9% had severe breathlessness, 16% experienced a decline in oxygen saturation of 3% or more during exercise testing, and 9% had finger clubbing.

## Factors associated with chronic lung disease

**Sociodemographic and pollution exposure factors.** The prevalence of CLD (basic definition) was higher in older adolescents aged 15–19 years than young adolescents 10–14 years or children <10 years (36% vs 19% vs 23% respectively), odds ratio (OR) 1.95, 95% confidence interval [CI] 1.01–3.75, p = 0.008 (Table 5). CLD was more prevalent among those currently in secondary school compared to lower levels (35% vs 16%. OR 2.92 [1.11–7.65]). We did not

**Table 4. Prevalence and severity of chronic lung disease in the study participants: Two clinical diagnostic approaches.**

| Characteristic (N = 320) | Detail | Chronic lung disease basic approach *(5 clinical feature)* | | Chronic lung disease enhanced approach *(add exercise limitation)* | |
|---|---|---|---|---|---|
| | | **Freq** | **%** | **Freq** | **%** |
| **No. of clinical features suggestive of CLD[a]** | **No. of symptoms/signs present** | | | | |
| ***Basic approach CLD:*** *Assess 5 symptoms/signs: chronic cough, breathlessness (MRC >2), SPO2 desaturation ≥3%, RR ≥99th centile, clubbing.* ***Enhanced approach CLD:*** *add moderate-to-severe exercise limitation.* | None | 117 | 36.6% | 104 | 32.5% |
| | One symptom or sign | 121 | 37.8% | 106 | 33.1% |
| | Two symptoms / signs | 59 | 18.4% | 71 | 22.2% |
| | Three symptoms / signs | 21 | 6.6% | 31 | 9.7% |
| | Four symptoms / signs | 2 | 0.6% | 7 | 2.2% |
| | Five symptoms / signs | 0 | 0.0% | 1 | 0.3% |
| | Six symptoms / signs | NA | NA | 0 | 0.0% |
| **Chronic Lung Disease** | **Classification** | | | | |
| | ***Present ($\geq$ 2 clinical features)*** | ***82*** | ***25.6%*** | ***110*** | ***34.4%*** |
| | Absent (0–1 clinical feature) | 238 | 74.4% | 210 | 65.6% |
| **Symptom phenotype among those with CLD** | **Specific symptom cough/breathless** | **(N = 82)** | | **(N = 110)** | |
| | Cough only | 10 | 12.2% | 13 | 11.8% |
| | Breathless only (MRC >2)[b] | 37 | 45.1% | 52 | 47.3% |
| | Cough and breathless | 30 | 36.6% | 30 | 27.3% |
| | Neither cough nor breathlessness | 5 | 6.1% | 15 | 13.6% |
| | *Total with cough* | 40 | 48.8% | 43 | 38.7% |
| | *Total with breathlessness* | 67 | 81.7% | 82 | 73.9% |
| **Severity of CLD symptoms/signs** | **Severity classification** | **(N = 82)** | | **(N = 110)** | |
| **Cough severity** | No cough | 42 | 51.2% | 67 | 60.9% |
| | Mild-to-moderate | 26 | 31.7% | 29 | 26.4% |
| | Severe | 14 | 17.1% | 14 | 12.7% |
| **Breathless severity by MRC dyspnoea score[b]** | MRC 1 (none) | 6 | 7.3% | 17 | 15.5% |
| | MRC 2 (mild) | 9 | 11.0% | 11 | 10.0% |
| | MRC 3 (moderate) | 58 | 70.7% | 72 | 65.5% |
| | MRC 4 (severe) | 9 | 11.0% | 10 | 9.1% |
| **SPO2 desaturation with exercise.[c]** *(5-criteria CLD N = 80. 6-criteria CLD N = 108)* | No change | 44 | 55.0% | 61 | 56.5% |
| | 1–2% drop | 19 | 23.8% | 30 | 27.8% |
| | 3% or more | 17 | 21.3% | 17 | 15.7% |
| **Finger clubbing** | Present | 7 | 8.5% | 10 | 9.1% |
| | Absent | 75 | 91.5% | 100 | 90.9% |
| **Six-minute walk test** | **Severity of exercise limitation** | | | **(N = 110)** | |
| **Percent of expected distance achieved.** *(N = 110)* | 80% and above (no limitation) | NA | NA | 19 | 17.3% |
| | 70–79% (mild limitation) | NA | NA | 40 | 36.4% |
| | 60–69% (moderate limitation) | NA | NA | 42 | 38.2% |
| | <60% (severe limitation) | NA | NA | 9 | 8.2% |

a-Chronic lung disease. b-Medical Research Council dyspnoea score. c-Difference in oxygen saturation measured by pulse oximetry pre- and post-exercise. NA-not applicable.

detect a significant difference in CLD prevalence between males and females. Parents/guardians with fewer years of education had higher odds of having a child with CLD compared to those with college level education (>12 years education); 1–8 years education OR 2.46 [1.10–5.50], and 9–12 years education OR 2.39 [1.09, 5.22], test for linear trend p = 0.040. There was a higher prevalence of CLD in participants living in one or two roomed houses compared to

**Table 5. Sociodemographic and environmental factors associated with chronic lung disease in the study population (unadjusted analysis).**

| Characteristic | Detail | N | CLD Present (N = 82) | | CLD Absent (N = 238) | | Odds Ratio | (95% CI) | p value | P across groups[c] |
|---|---|---|---|---|---|---|---|---|---|---|
| | | | Freq | % | Freq | % | | | | |
| **Sociodemography** | | | | | | | | | | |
| **Current age in years** | *Median (IQR)* | *320* | **14** | *(11,17)* | **12** | *(9,15)* | | | **0.0056** | |
| **Current age group** | <10 | 80 | 18 | 22.5% | 62 | 77.5% | Ref | | | |
| | 10 to 14 | 132 | 25 | 18.9% | 107 | 81.1% | 0.8 | (0.41, 1.59) | 0.532 | [0.023] |
| | 15 to 19 | 108 | 39 | 36.1% | 69 | 63.9% | **1.95** | **(1.01, 3.75)** | **0.046** | {0.008} |
| **Sex** | Male | 147 | 38 | 25.9% | 109 | 74.2% | 1.02 | (0.62, 1.69) | | |
| | Female | 173 | 44 | 25.4% | 129 | 74.6% | Ref | | 0.932 | |
| **Education level** | Pre-primary | 38 | 6 | 15.8% | 32 | 84.2% | Ref | | | |
| | Primary | 183 | 41 | 22.4% | 142 | 77.6% | 1.54 | (0.60, 3.94) | 0.367 | [0.006] |
| | Secondary | 99 | 35 | 35.4% | 64 | 64.7% | **2.92** | **(1.11, 7.65)** | **0.031** | {0.021} |
| **Toilet type** | Pit latrine | 225 | 63 | 27.9% | 163 | 72.1% | 1.53 | (0.85, 2.72) | 0.154 | |
| | Flush | 94 | 19 | 20.2% | 75 | 79.8% | Ref | | | |
| **No. rooms in home** | 1 to 2 | 196 | 60 | 30.6% | 136 | 69.4% | **2.05** | **(1.18, 3.55)** | **0.011** | |
| | >2 | 124 | 22 | 17.7% | 102 | 82.3% | Ref | | | |
| **Household density** | Mean (SD) | 320 | 2.4 | (1.6) | 2.0 | (1.6) | | | 0.055 | |
| **Caregiver education** | 0 to 8 years | 95 | 27 | 28.4% | 68 | 71.6% | **2.46** | **(1.10, 5.50)** | **0.028** | [0.054] |
| *N = 282* | 9 to 12 years | 115 | 32 | 27.8% | 83 | 72.2% | **2.39** | **(1.09, 5.23)** | **0.029** | {0.040} |
| | >12 years | 72 | 10 | 13.9% | 62 | 86.1% | Ref | | | |
| **Pollution Exposure** | | | | | | | | | | |
| **Use polluting cook fuel[a]** | Low | 159 | 48 | 30.2% | 111 | 69.8% | Ref | | | |
| | Moderate | 101 | 28 | 27.7% | 73 | 72.3% | 0.89 | (0.51, 1.54) | 0.67 | |
| | High | 60 | 6 | 10.0% | 54 | 90.0% | **0.26** | **(0.10, 0.64)** | **0.003** | {0.008} |
| **Car exhaust exposure** | Low | 77 | 16 | 20.8% | 61 | 79.2% | Ref | | | |
| | Moderate | 48 | 14 | 29.2% | 34 | 70.8% | 1.57 | (0.68, 3.60) | 0.287 | |
| | High | 195 | 52 | 26.7% | 143 | 73.3% | 1.39 | (0.73, 2.62) | 0.314 | {0.503} |
| **Any outdoor pollution exposure[b]** | Low | 42 | 5 | 11.9% | 37 | 88.1% | Ref | | | |
| | Moderate | 91 | 29 | 31.9% | 62 | 68.1% | **3.46** | **(1.23, 9.72)** | **0.018** | |
| | High | 132 | 31 | 23.5% | 101 | 76.5% | 2.27 | (0.82, 6.28) | 0.114 | |
| | Very high | 55 | 17 | 30.9% | 38 | 69.1% | **3.31** | **(1.11, 9.90)** | **0.032** | {0.068} |
| **COPD Assessment Test** | (HR-QoL) | | | | | | | | | |
| **CAT score** | *Median (IQR)* | *320* | **5** | *(0,11)* | **0** | *(0,3)* | | | **0.000** | |
| **CAT score categories** | 0–4 (higher QoL) | 251 | 38 | 15.10% | 213 | 84.90% | Ref | | | |
| | 5–9 (average QoL) | 33 | 19 | 57.60% | 14 | 42.40% | **7.61** | **(3.51,16.50)** | **0.000** | [0.000] |
| | 10+ (lower QoL) | 36 | 25 | 69.40% | 11 | 30.60% | **12.7** | **(5.79,28.0)** | **0.000** | {0.000} |

a Among 82 with CLD 6 (8%) had high pollutant cook fuel exposure; among 238 without CLD 53 (23%) had high polluting cook fuel exposure.

b Among 82 with CLD 17 (21%) had high outdoor pollution exposure; among 238 without CLD 38 (16%) had high outdoor pollution exposure.

c {} Chi-square test for homogeneity across groups. [] test for linear trend across groups]. CLD = Chronic lung disease, basic definition. OR = odds ratio.

CI = confidence interval. SD = standard deviation. IQR = inter-quartile range 25[th], 75[th] centile. HR-QoL = health-related quality of life assessed using COPD assessment test.

houses with more rooms (31% vs 18%, OR 2.05 [1.18–3.55]). Households of children with CLD were less likely to use polluting cooking fuel (7% vs 23%, OR 0.26, [0.10–0.64]). Those with high outdoor pollutant exposure had higher prevalence of CLD compared to those with low outdoor pollutant exposure (31% vs 12%, OR 3.31 [1.11–9.90]). Exposure to vehicle exhaust was similar in the two groups. Median CAT scores were higher among participants

with CLD compared to those without CLD (5 vs 0, respectively, p<0.001), suggesting that respiratory symptoms negatively impacted health-related quality of life among CAHIV with CLD.

**HIV disease factors at time of entry into HIV care and during follow-up.** Compared to participants without CLD those who had CLD: started ART later (median age 6.9 vs 5.1 years, p = 0.003), started cotrimoxazole prophylaxis later (median age 7 vs 5 years, p = 0.0001), initiated isoniazid preventive therapy later (median age 14 vs 12 years, p = 0.005) and had higher mean peak viral load (162,662 vs 66,782 copies/ml respectively, p = 0.045). WHO clinical stage and BMI at entry to care did not differ between the two groups (Table 6).

CAHIV with prior TB and/or pneumonia had higher prevalence of CLD compared to those with no prior lung infection (33% vs 19%, OR 2.04 [1.21–3.42]). Compared to CAHIV with no prior pneumonia or TB, those with history of having had both TB *and* pneumonia had the highest risk of CLD (41% vs 19%. OR 2.85 [1.12–7.24]), followed by those who had experienced TB only (33% vs 19%. OR 2.06 [0.91–4.67]) or pneumonia only (31% vs 19%. OR 1.85 [1.02–3.36]).

Current HIV disease status: Increasing viraemia was associated with increasing prevalence of CLD; 23%, 27% and 44% among those with viral load <50c/ml, between 50 to <400 and ≥400c/ml respectively (OR 2.7 [1.32–5.55], linear trend p = 0.008). Declining immunity appeared to be associated with a trend for increasing prevalence of CLD; 24.9%, 31.9% and 40.0% among those with CD4% of 25% and higher, 15–24% and <15% respectively (linear trend p = 0.161). Median current CD4 count was similar between participants with and without CLD (896 vs 900 cells/mm³). Participants with CLD had lower height for age z-score than those without CLD (median -0.92 vs -0.67, p = 0.042). Median BMI z-score and median weight for age z-score were not different between the two groups.

**Multivariable analysis of associated factors.** Ultimately eight factors entered the multivariable logistic regression model (Table 7, model 1). Age at ART initiation, at co-trimoxazole initiation and current age were correlated, therefore we retained age at ART initiation in the final model (Table 7, model 2). In the final model the following factors remained correlated with CLD: prior TB or pneumonia—adjusted odds ratio (aOR) 2.15, 95% CI 1.18–3.91, living in smaller houses of 1–2 rooms (aOR 1.95, [1.02–3.73]), CAT score (aOR 6.91 [3.66–13.03]) and use of polluting cooking fuel (aOR 0.35 [0.13–0.94]). Initiation of ART after age 5 years showed a trend for association with CLD (aOR 1.73 [0.93–3.20]).

## Discussion

Our study shows that chronic respiratory symptoms are common in CAHIV and affect all age-groups including young children. The commonest identified symptoms and signs were exertional breathlessness, chronic cough, tachypnoea and exercise limitation. A simple clinical diagnostic approach for CLD using presence of two or more of five clinical symptoms and signs identified one-quarter to have CLD, and addition of exercise limitation (reduced 6MWT distance) as a sixth criteria identified additional CLD cases. CLD was associated with prior TB or pneumonia, delayed initiation of ART, cotrimoxazole and isoniazid prophylaxis, detectable viraemia and sociodemographic factors, and had a negative impact on quality of life.

Our study included 80 children age <10 years, an age group for whom minimal data exists, and to our knowledge is the first study to evaluate CLD in African children below six years. We found that children had a similar prevalence to those aged 10–14 years. This was unexpected as younger children were born during the era of immediate start of ART at diagnosis of HIV during infancy, whereas those born before 2012 started ART only when they had more advanced clinical stage or depressed CD4 counts. This evidence suggests that even young

**Table 6. Clinical factors associated with chronic lung disease in the study population (unadjusted analysis).**

| Characteristic | Detail | N | CLD Present (N = 82) | | CLD Absent (N = 238) | | Odds ratio | 95% CI | p value | P across groups[a] |
|---|---|---|---|---|---|---|---|---|---|---|
| | | | Freq or median | % or IQR | Freq or median | % or IQR | | | | |
| **Age started ART in years** | <5 | 140 | 26 | 18.6% | 114 | 81.4% | Ref | | | |
| N = 317 | 5 to 9 | 126 | 37 | 29.4% | 89 | 70.6% | **1.82** | **(1.03, 3.23)** | **0.04** | [0.0173] |
| | 10+ | 51 | 19 | 37.3% | 32 | 62.7% | **2.60** | **(1.28, 5.29)** | **0.008** | [0.0046] |
| | *Median* | *317* | **6.9** | **4.1, 8.8** | **5.1** | **3.3, 7.2%** | | | *0.003* | |
| **Baseline WHO Stage** | 1–2 | 136 | 46 | 33.8% | 90 | 66.2% | Ref | | | |
| N = 252 | 3–4 | 116 | 29 | 25.0% | 87 | 75.0% | 0.65 | (0.38,1.13) | 0.128 | |
| **Prior Lung Infection** | TB and pneumonia | 22 | 9 | 40.9% | 13 | 59.1% | **2.85** | **1.12, 7.24** | **0.027** | |
| N = 308 | Pneumonia only | 84 | 26 | 31.0% | 58 | 69.0% | **1.85** | **1.02, 3.36** | **0.045** | {0.0413} |
| | TB only | 33 | 11 | 33.3% | 22 | 66.7% | 2.06 | 0.91, 4.67 | 0.083 | [0.0072] |
| | No TB no Pneumonia | 169 | 33 | 19.5% | 136 | 80.5% | Ref | | | |
| **Either TB or pneumonia** | Either TB or Pneumonia | 139 | 46 | 33.1% | 93 | 66.9% | **2.04** | **1.21, 3.42** | **0.007** | |
| N = 308 | No TB no Pneumonia | 169 | 33 | 19.5% | 136 | 80.5% | | | | |
| **Age started cotrimoxazole (years)** | <5 | 108 | 20 | 18.5% | 88 | 81.5% | | | | |
| N = 314 | 5 to <10 | 169 | 44 | 26.0% | 125 | 74.0% | 1.55 | (0.85,2.81) | 0.149 | {0.012} |
| | 10+ | 37 | 16 | 43.2% | 21 | 56.8% | **3.35** | **(1.49,7.55)** | **0.003** | [0.004] |
| | *Median age CTX* | *314* | **7** | **(4.5,9)** | **5** | **(3.6,7)** | | | *0.0001* | |
| **Age took IPT (years)** | <10 | 69 | 13 | 18.8% | 56 | 81.2% | Ref | | | |
| N = 273 | 10 to <14 | 133 | 27 | 20.3% | 106 | 79.7% | 1.09 | (0.52,2.29) | 0.805 | {0.004} |
| | 15+ | 71 | 28 | 39.4% | 43 | 61% | **2.81** | **(1.30,6.05)** | **0.009** | [0.0047] |
| | *Median age IPT* | *273* | **14** | **(11,16)** | **12** | **(9,14)** | | | *0.0052* | |
| **Highest recorded viral load (c/ml)** | *Mean (SD)* | *301* | *162,663* | *(586,012)* | *55,783* | *(239,296)* | | | *0.045* | |
| **Highest recorded viral load (c/ml)** | <10k | 217 | 53 | 24.4% | 164 | 75.6% | Ref | | | |
| N = 301 | 10k to <100k | 44 | 10 | 22.7% | 34 | 77.3% | 0.91 | (0.42,1.97) | 0.811 | {0.332} |
| | 100k+ | 40 | 14 | 35.0% | 26 | 65.0% | 1.67 | (0.81,3.42) | 0.164 | [0.245] |
| **Current Clinical Status** | | | | | | | | | | |
| **Recent viral load (c/ml)** | Undetectable or <50 | 254 | 58 | 22.8% | 196 | 77.2% | Ref | | | |
| | 50 to <400 | 30 | 8 | 26.7% | 22 | 73.3% | 1.22 | (0.52,2.91) | 0.639 | {0.021} |
| | 400+ | 36 | 16 | 44.4% | 20 | 55.6% | **2.7** | **(1.32,5.55)** | **0.007** | [0.0077] |
| **Current CD4%** | <15% | 10 | 4 | 40.0% | 6 | 60.0% | 2.01 | (0.55,7.36) | 0.291 | |
| N = 306 | 15–24% | 47 | 15 | 31.9% | 32 | 68.1% | 1.41 | (0.72,2.78) | 0.316 | {0.374} |
| | 25%+ | 249 | 62 | 24.9% | 187 | 75.1% | Ref | | | [0.161] |
| **Current CD4 count (cells/mm3)** | Median | 318 | 896 | 700, 1150 | 900 | 658, 1146 | | | 0.862 | |
| **Height for age** | Median HAZ | 318 | -0.915 | (-1.67,-0.22) | -0.67 | (-1.39,0.27) | | | **0.042** | |
| **Body mass index for age** | Median BMI-Z | 318 | -0.085 | (-1.1,0.49) | -0.375 | (-1.22,0.23) | | | 0.052 | |

CLD = chronic lung disease, basic definition. ART = antiretroviral therapy. WHO stage = World Health Organisation HIV clinical stage. TB = tuberculosis. IPT = isoniazid preventive therapy. OR = odds ratio. CI = confidence interval. SD = standard deviation. IQR = inter-quartile range 25th, 75th centile. QoL = quality of life. c/ml = copies per millilitre. a {} Chi-square test for homogeneity across groups. [] test for linear trend across groups.

**Table 7. Factors associated with chronic lung disease (multivariable analysis).**

| Model 1 (N = 297) | | | | |
|---|---|---|---|---|
| **Characteristic** | **Detail** | **Adjusted OR** | **95% CI** | **P value** |
| **Current age** | 10–14 vs <10yr | 1.29 | 0.52, 3.23 | 0.580 |
| | 15–19 vs <10yr | 1.49 | 0.51, 4.36 | 0.463 |
| **Age started cotrimoxazole** | 5+ vs <5yr | 0.63 | 0.17, 2.33 | 0.489 |
| **Age started ART** | 5+ vs <5yr | 2.12 | 0.64, 6.97 | 0.217 |
| **Prior TB or pneumonia** | Yes vs no | **1.99** | **1.07, 3.70** | **0.030** |
| **CAT score (HR-QoL)** | CAT score 5+ vs <5 | **7.45** | **3.84, 14.43** | **0.000** |
| **Use polluting cook fuel** | Moderate vs low | 0.84 | 0.42, 1.66 | 0.613 |
| | High vs low | 0.43 | 0.15, 1.19 | 0.105 |
| **No. rooms in house** | 1–2 vs >2 | 1.89 | 0.95, 3.76 | 0.071 |
| **Viral load (copies/ml)** | 50+ vs <50 | 1.30 | 0.61, 2.81 | 0.498 |
| Model 2 (N = 305) | | | | |
| **Characteristic** | **Detail** | **Adjusted OR** | **95% CI** | **P value** |
| **Age started ART** | *5+ vs <5 yr* | *1.73* | *0.93, 3.20* | *0.081* |
| **Prior TB or pneumonia** | Yes vs No | **2.15** | **1.18, 3.91** | **0.013** |
| **Use polluting cook fuel** | Moderate vs low | 0.75 | 0.39, 1.43 | 0.380 |
| | High vs low | **0.35** | **0.13, 0.94** | **0.037** |
| **No rooms in house** | 1–2 vs >2 | **1.95** | **1.02, 3.73** | **0.042** |
| **CAT score (HR-QoL)** | CAT score 5+ vs <5 | **6.91** | **3.66, 13.03** | **0.000** |

OR = odds ratio. CI = confidence interval. CAT = COPD assessment test. HR-QoL = health related quality of life.

children commenced on early ART are at risk of developing CLD. A more expected finding was that the older adolescents had the highest prevalence of CLD, likely a reflection of delayed ART initiation after progressive HIV disease and a higher incidence of lung infections over time.

Previous studies report chronic respiratory symptoms to be common in CAHIV, especially in those not established on ART or those who had delayed initiation [2, 15–17, 27, 28]. We found that most participants with cough also had increased sputum or chest congestion, as also seen in other studies [15, 16, 29]. Breathlessness was the most prevalent symptom and one-third reported moderate-to-severe exertional breathlessness during a 6MWT. Previous studies involving CAHIV in Zimbabwe and Malawi of a similar age range observed breathlessness in 11–34% [17, 27, 28, 30] and studies involving adolescents in Kenya and Zimbabwe found prevalence ranging 29–35% [15, 16]. In contrast, exertional breathlessness was uncommon in South African adolescents affecting only 3.4% [31]. Differences across settings may reflect cultural differences in perception of symptom severity or variation in approaches to assessing breathlessness. Researchers in Malawi and Zimbabwe used the New York Heart Association score, whereas Kenyan and South African studies used MRC dyspnoea score.

A quarter of participants reported a chest problem in the preceding year, and on more than one occasion in 14%. Prior studies in a similar age range as our study reported recurrent chest problems or antibiotics for respiratory problems in the preceding year in 21–22% of CAHIV [17, 27], with a higher prevalence (41%) in a study of adolescents [16]. This may be a useful additional criterion for clinical identification of CAHIV at risk of CLD. Respiratory symptoms such as tight chest, wheeze or chest pain were less common, as observed in other African paediatric studies [15–17, 27, 30, 31]. Tachypnoea at rest was common as in several previous studies that used a similar definition [32, 33], and more common than in studies that used a fixed cut-off of RR >25/min or >30/min regardless of age [16, 27, 28], suggesting that variation

may be due to categorisation used for tachypnoea. Finger clubbing, abnormal chest auscultatory findings and hypoxia at rest were uncommon and may indicate more advanced disease. Desaturation during exercise is a sign of impaired respiratory physiology likely to be more sensitive than hypoxia at rest [34]; it was observed in one-fifth of participants with CLD in our study and has been commonly reported elsewhere, especially in adolescents with HIV [15–17, 28, 32]. Early detection using readily available clinical features could potentially enable early treatment, and potentially reduce progression of CLD in this vulnerable population. Studies are needed that correlate clinical findings with objective evaluation of stage or severity of CLD.

There are limited investigations available in resource limited settings to objectively evaluate respiratory function, especially in young children. We performed 6MWT in all participants and one-fifth had moderate-to-severe exercise limitation. We found 6MWT to be feasible in children as young as three years with almost all completing and adhering to the protocol to objectively detect exercise induced desaturation and dyspnoea. In one previous African study that conducted 6MWT the youngest child was nine years (range 9–14 years), and participants achieved a lower median walk distance during 6MWT compared to ours [35]. A study in Zimbabwe involving children with and without HIV age 6–16 years used the incremental shuttle walk test which is a more aggressive exercise test than the 6MWT, and walk distance was significantly lower in children and adolescents with HIV compared to those without HIV [17]. In our study, adding exercise test distance as a sixth clinical feature to define CLD increased CLD prevalence in study population from 25.6 to 34.4%, an increase by one-third.

Despite challenges, a definition of CLD that facilitates clinical management but also standardises research evaluation and reporting is important, including for comparison between studies and over time in a range of settings [36]. Over the past decade, researchers have used varying approaches to define CLD in CAHIV initially using combinations of clinical symptoms and signs such as in the earliest proposed definition by Ferrand et al. in adolescents [16] which reported that 71% of adolescents had physician diagnosed CLD, higher than that observed in our cohort. Later in a younger cohort, Rylance et al. proposed a definition of any one of chronic cough, dyspnoea (MRC >1), or hypoxia at rest or with exercise which found, similar to our cohort, that 25% of participants had CLD [17]. Mwalukomo et al. proposed two clinical phenotypes in their study on CAHIV based on the commonest symptom (cough in 37.5%) and one common sign (hypoxia or desaturation with exercise in 38.8%) noting that only 22% had both cough and hypoxia [27]. Having cough and/or breathlessness with exertion identified around 90% of CLD in our cohort, but missed those in whom these symptoms are absent. The additional objective evaluations such as a brisk walk exercise test and pulse oximetry that we employed in our study are feasible at the primary health care level. Pulse oximetry is also increasingly available in lower-level health facilities. These findings provide additional evidence that may be helpful to inform clinical diagnostic protocols suitable for resource limited settings where there is minimal access to more costly diagnostics such as paediatric lung function testing and chest imaging.

### Factors associated with chronic lung disease

The identification of factors associated with CLD may help inform interventions for improved prevention and care [2, 10]. CLD in our cohort was associated with late initiation of ART and high viral load as previously described [2, 36]. We also found that previous pneumonia or tuberculosis were associated with CLD, also consistent with previous studies [2, 10, 12, 15, 36, 37]. The novel finding in our cohort was that earlier initiation of cotrimoxazole preventive treatment (CPT) and of isoniazid preventive treatment (IPT) were associated with a reduced risk of CLD. This may not be surprising as CPT and IPT prevent lung infections and disease

such as bacterial pneumonia, pneumocystis pneumonia and tuberculosis [2, 12, 38]. However, there is no previously published evidence that earlier CPT and/or IPT may reduce or prevent chronic lung sequelae in CAHIV. This has clear implications for prevention and management of CLD.

We noted that stunting was more common in CAHIV with CLD than those without but the implications for management are uncertain. Other studies have varying findings, hypoxia was associated with 3-fold increased odds of stunting in adolescents with HIV in Kenya [32], however prevalence of stunting was similar among CAHIV in Malawi with or without chronic cough [27]. Chronic respiratory morbidity can certainly negatively impact on growth in children by a number of mechanisms but whether nutritional support could reduce progression and severity of CLD is less certain. Finally, CLD had a negative impact on health-related quality of life in this cohort, as has been observed in Malawi where CAHIV with chronic cough had significantly lower scores on physical domains of the quality-of-life assessment tool [27]. Our findings underscore the importance of diagnosing HIV as early as possible in perinatally-infected children and providing comprehensive care with early ART initiation and prevention of respiratory infections to reduce the burden of CLD, and minimise morbidity associated with CLD.

Our participants had high exposure to air pollution both indoors and outdoors. Air pollution exposure causes chronic lung inflammation and progressive pathology regardless of HIV status. Exposure to indoor air pollution is a particular risk in low-income settings and is a potentially modifiable risk exposure that may be targeted as an intervention to reduce respiratory morbidity in this high-risk population [2, 10, 39, 40]. An unexpected finding was that families whose children had chronic respiratory symptoms were less likely to use polluting cooking fuel in the home than those whose children did not. It is plausible that when there is a child with respiratory symptoms in the home, a family would be more cautious and take measures to minimise smoke exposures inside the home. Given that most participants live in small houses of only one or two rooms they would have limited income to spend on the more expensive clean cooking fuels, and this is evidenced by the fact that some households with sick children did use polluting cooking fuel.

## Study limitations and strengths

Our study had several limitations. Firstly, our study population comprises children and adolescents receiving care at a tertiary referral hospital which may not represent the whole population of CAHIV with possible biases towards those with complications arising from severe HIV disease such as CLD. Secondly, survival bias may have impacted our findings given that this was an assessment of CAHIV still alive and in care at our hospital, and those who may have died due to more severe lung disease may be under-represented in this study. Thirdly, information on prior history of tuberculosis or pneumonia were largely from self-report rather than clinical records and so subject to recall bias. Fourthly, participants entered care during different years. Viral load and CD4 tests were not always routinely done at entry to care due to changing national guidelines over time so baseline results were not available for all participants. Finally, as previously highlighted, there are limitations inherent in definitions used to detect and evaluate CLD, especially less severe or minimally symptomatic disease. The strengths of our study include a large sample size and generation of insight on CLD in younger children who have been under-represented in previous studies. Our findings are possibly generalizable to CAHIV in care in tertiary care centres in low- and middle-income countries.

## Conclusions and recommendations

CLD affects one-third of CAHIV in an African setting, is prevalent in all age groups, and most are symptomatic with cough and/or exertional breathlessness. CLD significantly impacts their health-related quality of life, and is associated with prior lung infections, delayed ART, delayed prophylaxis against respiratory infections and socio-environmental factors. We conclude that a simple diagnostic approach using common respiratory symptoms and signs may be used to identify children and adolescents with CLD. There is need to increase awareness on the burden of CLD in the growing population of CAHIV. This evidence may be helpful to inform policy and guidelines for chronic lung disease in CAHIV, and to advocate for targeted structured clinical services specific to their needs in order to minimise morbidity and optimise their health and wellbeing.

## Supporting information

**S1 Checklist. PLOS' questionnaire on inclusivity in global research.**
(PDF)

**S1 Table. STROBE statement: Chronic lung disease in children and adolescents with HIV in Kenya.**
(PDF)

## Acknowledgments

The authors are indebted to "The Breathe Poa Study" children, adolescents and their parents/guardians, without their willing participation this research would not have been possible. We also appreciate the University of Nairobi and Kenyatta National Teaching and Referral hospital for providing a supportive environment to undertake this research. We would like to thank the entire clinical team and the data team for their dedicated diligent work throughout this research project. Special appreciation to Ms Electine Oyuga who coordinated all study logistics; to Daisy Chebet, Lynette Njeri and Eugene Makori for careful clinical assessments and exercise testing, to Christine Kundu for data management, to Peter Maleche Obimbo and Japheth Owiny for assistance in data organisation and analysis, and to Lynette Njeri for assistance in formatting of references.

## Author Contributions

**Conceptualization:** Elizabeth Maleche-Obimbo, Engi Attia.

**Formal analysis:** Elizabeth Maleche-Obimbo.

**Funding acquisition:** Elizabeth Maleche-Obimbo.

**Investigation:** Elizabeth Maleche-Obimbo.

**Methodology:** Elizabeth Maleche-Obimbo, Engi Attia, Fredrick Were, Walter Jaoko, Stephen M. Graham.

**Project administration:** Elizabeth Maleche-Obimbo.

**Resources:** Elizabeth Maleche-Obimbo, Engi Attia.

**Supervision:** Fredrick Were, Walter Jaoko, Stephen M. Graham.

**Validation:** Elizabeth Maleche-Obimbo.

**Visualization:** Elizabeth Maleche-Obimbo.

**Writing – original draft:** Elizabeth Maleche-Obimbo.

**Writing – review & editing:** Elizabeth Maleche-Obimbo, Engi Attia, Fredrick Were, Walter Jaoko, Stephen M. Graham.

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
