## [Decision Letter · Decision Letter 0]

7 Jul 2023

PONE-D-23-16528Prevalence, clinical presentation and factors associated with chronic lung disease among children and adolescents living with HIV in KenyaPLOS ONE

Dear Dr. Maleche-Obimbo,

Thank you for submitting your manuscript to PLOS ONE. After careful consideration, we feel that it has merit but does not fully meet PLOS ONE’s publication criteria as it currently stands. Therefore, we invite you to submit a revised version of the manuscript that addresses the points raised during the review process.

Please find below reviewr comments for your kind attention:

Reviewer 1

Abstract

Line 43: not clear what is the difference between exertional breathless and exercise limitation line 45.

Methods

Can the authors reference their definition of enhanced chronic lung disease (CLD)? Is their definition of enhanced CLD in literature?

Results

Table 1: what is the rationale of grouping children into <7 years, <7-9 years? Why not under 5 years category and 5-9 years old, for instance?

Line 247: what does the term “chest problem” refer to?

Table 4: CLD and CLD enhanced-are not those with breathlessness double counted with those with exercise limitation and vice versa?

Line 348 “took INH prophylaxis later” Please rephrase this statement to denote late initiation of INH prophylaxis

Line 353 “pneumonia” does this term perhaps refer to non-TB lower respiratory tract infection?

Can the term enhanced CLD be referenced from published literature?

Discussion

Line 399: exertional breathless vs exercise limitation. please see my comment above. Could the reduced distance walked during 6MWT in Table 4 be due to low oxygen saturation or breathlessness and hence this sub-group is already counted in the above?

Line 433 “chest problems” does this term refer to pain, infections, tight chest, etc?

Line 497 the authors state that stunting and CLD implication of management is uncertain. Is this true, considering stunting denotes chronic malnutrition and nutrition impacts lung growth; height for instance impacts lung function.

References

Refs 11 and 39 are the same, Verwey, et al.

Reviewer 2

1. This is an interesting study addressing a topic with clear scientific value. Due to the period of the study (covid-19), with the exception of the standardized six-minute walk test (6MWT) being carried out outdoors for safety precautions, no other implications of covid-19 are mentioned in the study results. Evaluate and discuss implications of covid-19 on study results, if applicable.

2. There is need to provide a study profile including n of screened participants, n of excluded and n of participants included in the different evaluations for a understanding of the reader.

3. Actors should mention in the article the definition of outdoor pollution exposure definition.

4. Although the article collected information on the level of education, this seems to me to be decisive for older adolescents. In younger children, the child's health is mostly determined by the caregiver's education and not the child's. A limitation of the study is the fact that there is no information on the primary caregiver's education and also no information on who the questionnaire was administered to.

A rebuttal letter that responds to each point raised by the academic editor and reviewer(s). You should upload this letter as a separate file labeled 'Response to Reviewers'. A marked-up copy of your manuscript that highlights changes made to the original version. You should upload this as a separate file labeled 'Revised Manuscript with Track Changes'. An unmarked version of your revised paper without tracked changes. You should upload this as a separate file labeled 'Manuscript'.

If applicable, we recommend that you deposit your laboratory protocols in protocols.io to enhance the reproducibility of your results. Protocols.io assigns your protocol its own identifier (DOI) so that it can be cited independently in the future. For instructions see: https://journals.plos.org/plosone/s/submissionguidelines#loc-laboratory-protocols. Additionally, PLOS ONE offers an option for publishing peer-reviewed Lab Protocol articles, which describe protocols hosted on protocols.io. Read more information on sharing protocols at https://plos.org/protocols?utm_medium=editorialemail&utm_source=authorletters&utm_campaign=protocols. 

We look forward to receiving your revised manuscript.

Kind regards,

Judith Kose, M.D.

Academic Editor

PLOS ONE

Journal Requirements:

Additional Editor Comments:

Thank for submitting your manuscript to PLOS ONE. This is a very well written paper and addresses a key topic. The study findings are very relevant to low-to-middle-income settings where HIV-related lung disease is common. We have reviewed the manuscript. Please find below comments for your attention

Reviewer 1

Abstract

Line 43: not clear what is the difference between exertional breathless and exercise limitation line 45.

Methods

Can the authors reference their definition of enhanced chronic lung disease (CLD)? Is their definition of enhanced CLD in literature?

Results

Table 1: what is the rationale of grouping children into <7 years, <7-9 years? Why not under 5 years category and 5-9 years old, for instance?

Line 247: what does the term “chest problem” refer to?

Table 4: CLD and CLD enhanced-are not those with breathlessness double counted with those with exercise limitation and vice versa?

Line 348 “took INH prophylaxis later” Please rephrase this statement to denote late initiation of INH prophylaxis

Line 353 “pneumonia” does this term perhaps refer to non-TB lower respiratory tract infection?

Can the term enhanced CLD be referenced from published literature?

Discussion

Line 399: exertional breathless vs exercise limitation. please see my comment above. Could the reduced distance walked during 6MWT in Table 4 be due to low oxygen saturation or breathlessness and hence this sub-group is already counted in the above?

Line 433 “chest problems” does this term refer to pain, infections, tight chest, etc?

Line 497 the authors state that stunting and CLD implication of management is uncertain. Is this true, considering stunting denotes chronic malnutrition and nutrition impacts lung growth; height for instance impacts lung function.

References

Refs 11 and 39 are the same, Verwey, et al.

Reviewer 2

1. This is an interesting study addressing a topic with clear scientific value. Due to the period of the study (covid-19), with the exception of the standardized six-minute walk test (6MWT) being carried out outdoors for safety precautions, no other implications of covid-19 are mentioned in the study results. Evaluate and discuss implications of covid-19 on study results, if applicable.

2. There is need to provide a study profile including n of screened participants, n of excluded and n of participants included in the different evaluations for a understanding of the reader.

3. Actors should mention in the article the definition of outdoor pollution exposure definition.

4. Although the article collected information on the level of education, this seems to me to be decisive for older adolescents. In younger children, the child's health is mostly determined by the caregiver's education and not the child's. A limitation of the study is the fact that there is no information on the primary caregiver's education and also no information on who the questionnaire was administered to.

Reviewers' comments:

Reviewer's Responses to Questions

**Comments to the Author**

1. Is the manuscript technically sound, and do the data support the conclusions?

Reviewer #1: Yes

Reviewer #2: Yes

2. Has the statistical analysis been performed appropriately and rigorously? 

Reviewer #1: Yes

Reviewer #2: Yes

3. Have the authors made all data underlying the findings in their manuscript fully available?

Reviewer #1: Yes

Reviewer #2: Yes

4. Is the manuscript presented in an intelligible fashion and written in standard English?

Reviewer #1: Yes

Reviewer #2: Yes

5. Review Comments to the Author

Reviewer #1: This is a well-written article on a very important topic. 

Abstract

Line 43: not clear what is the difference between exertional breathless and exercise limitation line 45.

Methods

Can the authors reference their definition of enhanced chronic lung disease (CLD)? Is their definition of enhanced CLD in literature?

Results

Table 1: what is the rationale of grouping children into <7 years, <7-9 years? Why not under 5 years category and 5-9 years old, for instance?

Line 247: what does the term “chest problem” refer to?

Table 4: CLD and CLD enhanced-are not those with breathlessness double counted with those with exercise limitation and vice versa?

Line 348 “took INH prophylaxis later” Please rephrase this statement to denote late initiation of INH prophylaxis

Line 353 “pneumonia” does this term perhaps refer to non-TB lower respiratory tract infection?

Can the term enhanced CLD be referenced from published literature?

Discussion

Line 399: exertional breathless vs exercise limitation. please see my comment above. Could the reduced distance walked during 6MWT in Table 4 be due to low oxygen saturation or breathlessness and hence this sub-group is already counted in the above?

Line 433 “chest problems” does this term refer to pain, infections, tight chest, etc?

Line 497 the authors state that stunting and CLD implication of management is uncertain. Is this true, considering stunting denotes chronic malnutrition and nutrition impacts lung growth; height for instance impacts lung function.

References

Refs 11 and 39 are the same, Verwey, et al.

Reviewer #2: 1. This is an interesting study addressing a topic with clear scientific value. Due to the period of the study (covid-19), with the exception of the standardized six-minute walk test (6MWT) being carried out outdoors for safety precautions, no other implications of covid-19 are mentioned in the study results. Evaluate and discuss implications of covid-19 on study results, if applicable.

2. There is need to provide a study profile including n of screened participants, n of excluded and n of participants included in the different evaluations for a understanding of the reader.

3. Actors should mention in the article the definition of outdoor pollution exposure definition.

4. Although the article collected information on the level of education, this seems to me to be decisive for older adolescents. In younger children, the child's health is mostly determined by the caregiver's education and not the child's. A limitation of the study is the fact that there is no information on the primary caregiver's education and also no information on who the questionnaire was administered to.

6. PLOS authors have the option to publish the peer review history of their article (what does this mean?). If published, this will include your full peer review and any attached files.

Reviewer #1: No

Reviewer #2: No

---

## [Author Response · Author response to Decision Letter 0]

20 Jul 2023

Academic Editor Requirements:

• A rebuttal letter that responds to each point raised by the academic editor and reviewer(s). You should upload this letter as a separate file labeled 'Response to Reviewers'. DONE

• A marked-up copy of your manuscript that highlights changes made to the original version. You should upload this as a separate file labeled 'Revised Manuscript with Track Changes'. DONE

• An unmarked version of your revised paper without tracked changes. You should upload this as a separate file labeled 'Manuscript'. DONE 

Journal Requirements:

Response: We have formatted our manuscript to conform to PLOS ONE’s style requirements.

Response: The lead researchers and co-investigators are Kenyan citizens and conducted research within our own country Kenya. The policy therefore may not really apply to us, nevertheless I have filled this form as requested. 

Response: We have included this information in the Methods section of the manuscript.

Response: We have done so. One new reference was added in methods section (ref 22). One reference citation which we had inadvertently duplicated in the ref list (was cited twice as ref 11 and ref 39) this was corrected. Total number of references remains 41. 

New reference no. 22:

ATS committee on proficiency standards for clinical pulmonary function laboratories. ATS statement: guidelines for the six-minute walk test. Am J Respir Crit Care Med. 2002 Jul 1;166(1):111-7. doi: 10.1164/ajrccm.166.1.at1102. Erratum in: Am J Respir Crit Care Med. 2016 May 15;193(10):1185. PMID: 12091180.

Duplicated reference (pointed out by the second reviewer) now cited once as ref no. 11:

Verwey C, Gray DM, Dangor Z, Ferrand RA, Ayuk AC, Marangu D, et al. Bronchiectasis in African children: challenges and barriers to care. Front Pediatr. 2022 Jul 25; 10:954608. doi: 10.3389/fped.2022.954608. PMID: 35958169; PMCID: PMC9357921

 

Responses to Reviewer Comments

Reviewer 1

Abstract

Line 43: not clear what is the difference between exertional breathless and exercise limitation line 45.

Response: Exertional breathlessness – this was the response to the question “do you get short of breath when you climb stairs or when walking uphill?”. This question was posed to the child/adolescent during the interview. 

Exercise limitation – this refers to limitation in physical activity. This was assessed using a standardized exercise test – the six-minute walk test commonly used to assess if an individual has limitation in physical activity, or exercise limitation (Ref 22). A participant walks as fast as they can on flat ground for six minutes, and the total distance walked is recorded. The distance achieved is then compared against a benchmark of the expected distance that a healthy child of similar height, weight and age should walk, and reported as a percentage of predicted distance (ref 25). Details on the methods and classification of exercise limitation are provided in methods section of the manuscript (manuscript with tracked changes: p6, lines 135-137, p8 lines 170-172). 

Methods

Can the authors reference their definition of enhanced chronic lung disease (CLD)? Is their definition of enhanced CLD in literature?

Response: There are no standardized definitions of chronic lung disease for HIV infected children and adolescents, and researchers have used varying definitions tailored to their specific studies and clinical settings, these studies are mentioned in the introduction section of the manuscript (line 89, refs 10-11, 15-21). They are further mentioned in the discussion section (p31, lines 548 – 554, refs 17, 18).

We examined the various definitions in published literature in CAHIV on this subject, when considering which symptoms and signs to use in the CLD definition in our study. The commonest symptoms suggestive of CLD used across studies are chronic cough, and breathlessness. Common physical examination signs suggestive of CLD used across studies include finger clubbing, and abnormal oxygen saturation were also popular. Some studies subjected participants to exercise but the outcome parameters used were exercise induced breathlessness, and hypoxia/desaturation, and the exercise tests used differed from ours (p20 – 30, lines 502-511, 519 – 527, 538-542, refs 16-18, 28-32). We found one study by Githinji et al on CAHIV that conducted the six-minute walk test and measured walk distance as an outcome of interest, however they did not use this parameter as part of diagnosis for CLD (p30,line 536-538, ref 36). They conducted extensive lung function testing as their main diagnostic tool for diagnosis of CLD. 

Results

Table 1: what is the rationale of grouping children into <7 years, <7-9 years? Why not under 5 years category and 5-9 years old, for instance? 

Response: We noted from review of literature that previous researchers had excluded children below 7 years – so our study is the first to provide insight on this problem among children <7 years. For this reason we thought it useful to display the number of children of age-group <7 yr that we enrolled in this study. We have added a footer to table 1 stating: “Additional detail on children <10 yrs: 14 (4.4%) were age <5 yr; 66 (20.6%) were age 5-9 yr.” (p12, line 261).

Line 247: what does the term “chest problem” refer to?

Response: The participants were asked “In the last 12 months have you had a chest problem, outside of a common cold?” 

Further details of the chest problem were also asked – specifically we asked if they had experienced congested chest, tight chest, or chest pain in the preceding 12 months. 

Table 4: CLD and CLD enhanced-are not those with breathlessness double counted with those with exercise limitation and vice versa?

Response: Thank you for this thoughtful query. We also considered this when examining our findings. In this study population some children experienced neither symptom, some experienced one and not the other, whereas some experienced both breathlessness and exercise limitation. The latter group would then be identified with two symptoms suggestive of CLD, but those with only one of the two criteria would not be identified as CLD based on these two specific clinical features. It is unclear why some breathless children achieved normal walk distance and others did not, and vice versa. We do note that the enhanced definition appears to capture additional subjects who were not identified using the basic CLD approach. 

There is indeed need for further research to better elucidate the pathophysiology leading to breathlessness and to exercise limitation, and the interaction between the two, as well as other clinical symptoms and signs suggestive of chronic lung pathology.

Line 348 “took INH prophylaxis later” Please rephrase this statement to denote late initiation of INH prophylaxis

Response: This has been rephrased. (p21, Line 420)

Line 353 “pneumonia” does this term perhaps refer to non-TB lower respiratory tract infection?

Response: The participants were asked if “Have you ever had pneumonia or chest infection other than TB”. In general any acute chest infection is described as “pneumonia” by non-medical persons in our setting, and is not distinguished from bronchiolitis or bronchitis. 

Can the term enhanced CLD be referenced from published literature?

Response: The term is “enhanced definition of CLD” rather than “enhanced CLD”. This is a term specific to this study. We have given more detailed response to this in the earlier comment raised by the reviewer, please see earlier section. 

Discussion

Line 399: exertional breathless vs exercise limitation. Please see my comment above. Could the reduced distance walked during 6MWT in Table 4 be due to low oxygen saturation or breathlessness and hence this sub-group is already counted in the above?

Response: Thank you for your thoughtful analysis of the various scenarios. We concur that exercise induced symptoms may then lead to a child slowing down and this in turn lead to reduced distance walked. 

We tried to be conservative in the definition of exercise limitation and used a cut-off of “achieved <70% of predicted distance” or moderate limitation as opposed to <80% of predicted (includes those with mild exercise limitation) in order to minimize over-calling exercise limitation (p8, line 186 – 191). 

In this study population all the 17 participants who experienced oxygen desaturation of 3% or more were identified in the basic CLD group, in other words all 17 had at least one additional symptom/sign (cough, breathlessness, tachypnoea or finger clubbing) regardless of distance walked. 

Some children in our study reported being short of breath during the walk, but it did not limit their walk distance, and this group of children the enhanced CLD approach did not re-categorize them as CLD if they had none of the other four symptoms/signs. It is possible that children who got breathless but had no other respiratory symptoms or signs (suggestive of healthy respiratory system) and had normal walk distance were simply a bit physically unfit. Such a child would not be categorized as having CLD either with the basic definition or the enhanced definition. 

The uestionn that you raise points to a need for further research that would provide insight into the pathologic changes and pathophysiology of CLD, and the relative importance of each specific respiratory symptom and sign in diagnosing CLD in these vulnerable CAHIV. We attempted to articulate some of these issues in the discussion of these findings. 

Line 433 “chest problems” does this term refer to pain, infections, tight chest, etc?

Response: The participants were asked “In the last 12 months have you had a chest problem, outside of a common cold?” 

They were also asked what type of chest problem they had experienced – specifically we asked if they had experienced congested chest, tight chest, or chest pain in the preceding 12 months. 

Line 497 the authors state that stunting and CLD implication of management is uncertain. Is this true, considering stunting denotes chronic malnutrition and nutrition impacts lung growth; height for instance impacts lung function.

Response: Thank you for your thoughtful queries to this statement. Given that the height and CLD assessment were done at the same point in time, it is not possible to know which occurred before the other. We do concur with your observation that stunting may denote chronic malnutrition and nutrition impacts lung growth. However we were also cognissant of the fact that in these CAHIV causes of stunting may be multifactorial, due to chronic inadequate nutrition, or due to other HIV-related morbidity, and in CLD may be due to chronic sub-optimal oxygenation, or recurrent lung infections due to damaged lungs. For this reason we were cautious in our interpretation of this finding and its implications for management.

References

Refs 11 and 39 are the same, Verwey, et al.

Response: Thank you for pointing this out, we have deleted the duplicate ref 39, and re-aligned the references as appropriate.

 

Reviewer 2

1. This is an interesting study addressing a topic with clear scientific value. Due to the period of the study (covid-19), with the exception of the standardized six-minute walk test (6MWT) being carried out outdoors for safety precautions, no other implications of covid-19 are mentioned in the study results. Evaluate and discuss implications of covid-19 on study results, if applicable.

Response: Thank you for this comment, it is an important factor to consider. We have added a statement on effect of covid-19 pandemic at the beginning of the results (p10, lines 232 - 234). The statement reads as follows:

“Due to the covid-19 pandemic active enrolment was stopped on 18th March 2020 after enrolling the first 14 participants, and restarted eight months later on 25th November 2020 with covid-19 precautions in place for participants and staff”. 

The major implication for the study is that enrolment took longer than initially anticipated, however we were still able to achieve the targeted sample size required for the study, and participants were able to complete the interview, physical examination and exercise testing since the latter was done outdoors. By the time we restarted the study parents were generally continuing with normal work and other activities, children were on school vacation and interacting freely within their neighborhoods. Very few study children (only two that we are aware of) were diagnosed with covid-19 during the study period, however access to testing was low due to the high cost of the test, and the low socio-economic status of our study population. The two children had mild covid-19 symptoms and recovered fully. 

We see no additional significant implication of covid-19 on the study results. 

2. There is need to provide a study profile including n of screened participants, n of excluded and n of participants included in the different evaluations for a understanding of the reader.

Response: Thank you for bringing this to our attention. 

Details of screening, exclusion, enrolment and completion of clinical evaluations have been added in the first paragraph of the results (p10, line 229 – 232, 234 - 244). 

Briefly we screened 326, two were ineligible as they were from a children’s home, and four potentially eligible children declined to participate due to non-interest. Of the 320 enrolled, all completed the interview, all underwent physical examination and pulse oximetry assessment, and 312 completed the six-minute walk exercise test. 

There was very high acceptance by study participants, perhaps they felt it was beneficial to know the status of their (their child’s) lungs. The interviews and procedures were carried out largely by clinicians and nurses who had cared for the children and adolescents for many years, so they had good rapport with them. 

3. Actors should mention in the article the definition of outdoor pollution exposure definition.

Response: We have added details on definition of outdoor pollution exposure in the methods section of the manuscript. (p7, line 157 – 160)

4. Although the article collected information on the level of education, this seems to me to be decisive for older adolescents. In younger children, the child's health is mostly determined by the caregiver's education and not the child's. A limitation of the study is the fact that there is no information on the primary caregiver's education and also no information on who the questionnaire was administered to.

Response: Thank you for this insightful comment, we concur it is an important characteristic to evaluate. We had omitted this data due to the size of the manuscript. We have now added the caregiver characteristics in table 1 (page 12). Caregiver education data available for 282 (88%) of 320 participants. We have added a summary of this information in the narrative. (p12-13, lines 264 – 269). Association between caregiver education and CLD is reported on Table 5 (p20-21, lines 401 – 407. Table 5 p23). Caregivers with fewer years of education (1-8 yrs) had higher odds of their child having CLD thank those with >12 years education (crude OR 2.46, 95% CI 1.10 – 5.50, p=0.028). 

Due to missing data on >10% of participants, per the a-priori multivariable model statistical plan this was not added to the adjusted analysis. However caregiver education is associated with socio-economic status, which was represented in the final multivariable models by the variable number of rooms in the home, which was an independent predictor of CLD in this study population. 

 

Comments to the Author

1. Is the manuscript technically sound, and do the data support the conclusions?

Reviewer #1: Yes

Reviewer #2: Yes

2. Has the statistical analysis been performed appropriately and rigorously? 

Reviewer #1: Yes

Reviewer #2: Yes

3. Have the authors made all data underlying the findings in their manuscript fully available?

Reviewer #1: Yes

Reviewer #2: Yes

4. Is the manuscript presented in an intelligible fashion and written in standard English?

Reviewer #1: Yes

Reviewer #2: Yes

Concluding Remarks.

The authors of the manuscript titled “Prevalence, clinical presentation and factors associated with chronic lung disease among children and adolescents living with HIV in Kenya (PONE-D-23-16528)” appreciate the careful review and comments by the two reviewers, and by the academic editor. We do hope that we have responded adequately to the comments of the reviewers, and that we have addressed all the requirements of the editors and journal. 

Elizabeth Maleche-Obimbo

---

## [Editor Report · Decision Letter 1]

25 Jul 2023

Prevalence, clinical presentation and factors associated with chronic lung disease among children and adolescents living with HIV in Kenya

PONE-D-23-16528R1

Dear Dr. Obimbo,

We’re pleased to inform you that your manuscript has been judged scientifically suitable for publication and will be formally accepted for publication once it meets all outstanding technical requirements.

Kind regards,

Judith Kose, M.D.

Academic Editor

PLOS ONE

Additional Editor Comments (optional):

Dear Dr Obimbo,

I hope this message finds you well.

Congratulations on your outstanding work and the significant contributions your research brings to the scientific community. Your manuscript showcases a commendable level of rigor, clarity, and originality, making it a valuable addition to our journal.

The revisions you made to the manuscript were thorough and addressed all the concerns raised during the peer review process. Your commitment to improving the manuscript has resulted in a much stronger and polished piece of work.

Thank you for choosing PLOS ONE as the platform to disseminate your valuable research.

Best regards,

Dr Kose

Academic Editor

PLOS ONE

---

## [Editor Report · Acceptance letter]

31 Jul 2023

PONE-D-23-16528R1 

Prevalence, clinical presentation and factors associated with chronic lung disease among children and adolescents living with HIV in Kenya 

Dear Dr. Maleche-Obimbo:

I'm pleased to inform you that your manuscript has been deemed suitable for publication in PLOS ONE. Congratulations! Your manuscript is now with our production department. 

Kind regards, 

on behalf of

Dr. Judith Kose 

Academic Editor

PLOS ONE